# DynamicRAG: Leveraging Outputs of Large Language Model as Feedback for Dynamic Reranking in Retrieval-Augmented Generation

**Jiashuo Sun, Xianrui Zhong, Sizhe Zhou, Jiawei Han**[*]
University of Illinois Urbana-Champaign
{jiashuo5, hanj}@illinois.edu

## Abstract

Retrieval-augmented generation (RAG) systems combine large language models (LLMs) with external knowledge retrieval, making them highly effective for knowledge-intensive tasks. A crucial but often under-explored component of these systems is the reranker. Since irrelevant documents in RAG systems can mislead the generator, the reranker plays a vital role in refining retrieved documents to enhance generation quality and explainability. However, it is challenging to determine the appropriate number of documents ($k$) that the reranker should select: too few may result in missing critical information, while too many introduce noise and inefficiencies. Although recent studies have explored LLM-based rerankers, they primarily leverage internal model knowledge and overlook the rich supervisory signals that LLMs can provide, such as using response quality as feedback for optimizing reranking decisions. In this paper, we propose DynamicRAG, a novel RAG framework where the reranker dynamically adjusts both the order and number of retrieved documents based on the query. We model the reranker as an agent optimized through reinforcement learning (RL), using rewards derived from LLM output quality. Across seven knowledge-intensive datasets, DynamicRAG demonstrates superior performance, achieving state-of-the-art results among models of same parameter sizes.

## 1 Introduction

Retrieval-augmented generation (RAG) systems have emerged as a powerful approach for combining the strengths of large language models (LLMs) with external knowledge retrieval. This integration has proven highly effective for addressing knowledge-intensive tasks and incorporating up-to-date information into LLMs, leading to notable performance improvements [17, 24, 14]. RAG systems often suffer from two critical challenges: misleading irrelevant retrieved documents that can distort the generation process, and the 'lost-in-the-middle' phenomenon where important information gets buried within long lists of retrieved candidates. A crucial, yet often underappreciated, component that addresses these issues is the reranker, which assesses the relevance of retrieved documents. The reranker is critical for improving the quality of generated text and enhancing explainability, thereby serving as an indispensable part of the RAG framework [33, 42]

In RAG systems, the reranker's primary role is to refine the Top-$N$ documents retrieved by the retriever, selecting the $k$ most relevant ones to enhance the answer quality. However, determining the optimal k remains a challenging problem, as highlighted in previous studies [36, 62]. A $k$ that is too small risks omitting critical information, leading to degraded generation quality, while a larger $k$ may introduce irrelevant content, increasing noise and potentially misleading the generator.

---

[*]Corresponding author.

39th Conference on Neural Information Processing Systems (NeurIPS 2025).

Furthermore, incorporating excessively long contexts can reduce both efficiency and effectiveness, further complicating the balance between relevance and performance in RAG systems. Striking the right balance requires adaptive strategies that can dynamically adjust $k$ based on query complexity and document diversity, as shown in Figure 1.

Recent work has demonstrated the effectiveness of LLM-based rerankers [9, 53, 26, 23, 30], which leverage large language models' capabilities to assess document relevance and improve ranking quality. Other works have also used the understanding capabilities of LLMs through sliding window mechanisms to achieve optimal re-ranking results [53, 5]. While these studies demonstrate the effectiveness of LLM-based rerankers, they typically rely on fixed document selection thresholds ($k$) and fail to dynamically adapt to varying query complexity and retrieval quality. Existing approaches primarily exploit LLMs' internal knowledge to score documents independently, overlooking a key insight: the actual generation quality when using different numbers of documents provides direct feedback about the optimal $k$. This natural reward signal enables us to apply reinforcement learning, where the LLM's generation quality serves as the reward for selecting the right number of documents. This supervisory signal—derived from the downstream generation task itself—offers a more principled approach to document selection than static thresholds or isolated relevance scoring. For instance, the quality of an LLM's response—given a query and a ranked set of documents—serves as a direct indicator of document relevance.

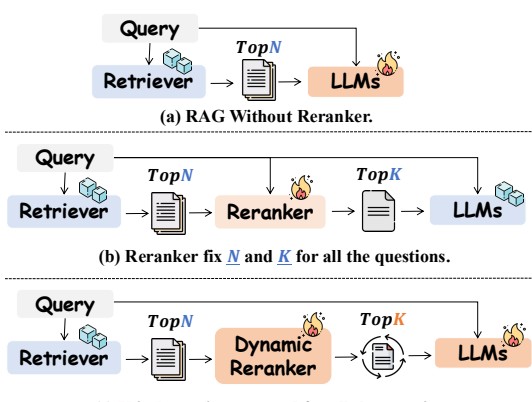

Figure 1: Illustration of our dynamic reranker framework. (a) It represents a RAG system without a reranker, where the system primarily focuses on training LLMs. (b) It represents a RAG system with a reranker, where the reranker is trained to filter the Top-$N$ documents to a fixed Top-$K$, which remains constant for all queries. (c) In contrast, it represents our dynamic reranker, where both the reranker and the generator are trained simultaneously. The dynamic reranker adapts to the difficulty of each query by dynamically determining the value of $k$.

Based on these insights, we propose DynamicRAG, a novel RAG framework where the reranker dynamically adjusts both the order and number of retrieved documents based on the query. In DynamicRAG, the reranker is modeled as an agent optimized through reinforcement learning (RL), with rewards derived from the evaluated quality of LLM outputs. The entire training process consists of two stages. First, we adopt behavior cloning by collecting expert trajectories and training the reranker via supervised fine-tuning (SFT). This provides the reranker with a basic understanding of the dynamic reranking task while reducing the complexity of the action space. Second, we treat the generator as an interactive environment that provides feedback, enabling the reranker to explore, collect trajectories, and update itself through reinforcement learning.

We comprehensively evaluate DynamicRAG on knowledge-intensive tasks across seven datasets, including general QA (NQ [25], TriviaQA [19]), multi-hop reasoning (HotpotQA [59], 2WikimQA [15]), long-form generation (ASQA [51], ELI5 [10]), and fact verification (FEVER [55]). Additionally, we assess the recall results of DynamicRAG's reranker on NQ and HotpotQA. Experimental results show that DynamicRAG significantly outperforms existing fine-tuned and prompting-based approaches, achieving state-of-the-art (SOTA) performance among models of same size while requiring substantially less training data.

## 2 DynamicRAG

In this section, we propose DynamicRAG. Unlike traditional RAG systems that rely on static ranking methods, DynamicRAG introduces a dynamic reranking mechanism and leverages feedback from LLM output to further refine the reranker, thereby achieving overall optimization of the RAG system.

The DynamicRAG framework consists of three key components: (1) a frozen retriever, (2) a trainable dynamic reranker, and (3) a trainable generator that is optimized to effectively leverage the reranker's dynamically selected $k$ documents. The retriever retrieves relevant documents from a large corpus, while the reranker dynamically determines both the order and the number of documents to be passed to the generator to produce an answer. The generator then produces the final output based on the reranker's selected documents. By iteratively training the reranker and generator, DynamicRAG achieves improvements in the overall efficiency and effectiveness of the RAG system.

## 2.1 Dynamic Reranking

Traditional reranking approaches rely on static ranking models that determine the relevance of retrieved documents independently of the generation task. These models typically operate within a fixed-length input framework, where a list comprising $n$ documents serves as the input, and the output is a reordered sequence containing the same $n$ documents. This inherent limitation prevents these static models from dynamically adapting to the specific needs of the generation process, particularly in terms of the number and arrangement of the selected documents. In contrast, DynamicRAG uses feedback from the generator to guide the reranking process, allowing it to dynamically adjust both the order and the number of documents selected. This enables the reranker to optimize the input of the generator, maximizing the likelihood of producing high-quality output (In this paper, we only consider the list-wise ranking).

To formalize this operation, the reranking process can be expressed as a single function that directly outputs a reordered subset of the initially retrieved documents. Given a query $\mathbf{q}$ and a set of retrieved documents $D = \{D_1, D_2, \ldots, D_k\}$, the reranker computes:

$$\hat{D} = \texttt{Reranker}_{\theta_r}(\mathbf{q}, D), \quad (1)$$

where $\hat{D} \subset D$ is the selected and reordered subset of documents, and $\theta_r$ represents the parameters of the reranker model. This formulation encapsulates both the scoring and selection processes, producing a subset $\hat{D}$ with dynamically adjusted order and size. This dynamic adjustment enables the model to exhibit greater flexibility in accommodating diverse queries and the specific requirements of the generation task.

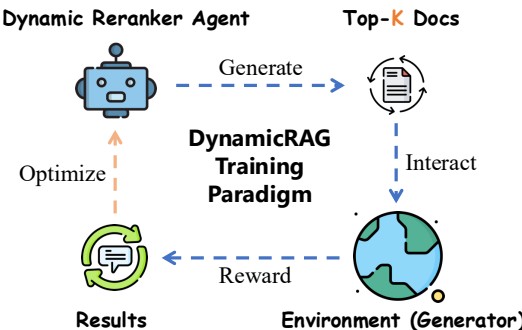

Figure 2: Illustration of the training paradigm for our method. We treat Dynamic Reranker as an Agent, which interacts with the Environment, generating Top-K docs and receiving rewards to optimize itself.

## 2.2 Training with Interacting

In this section, we outline the training process for the Reranker, where the architectural overview of which is schematically depicted in Figure 2. The training begins with behavior cloning, which equips the Reranker with the basic ability to adjust both the order and the number of selected documents. Building upon this foundational model, we further refine the Reranker's performance through interactions with the environment. This interaction allows the Reranker to collect feedback from multiple trajectories and enhance its decision-making policy. The complete training procedure is illustrated in Figure 3 and is presented in detail in Algorithm 1.

### 2.2.1 Behavioral Cloning with Expert Trajectories

Behavioral cloning [58, 63] is used to supervised fine-tuning the Reranker by mimicking expert trajectories, allowing the model to learn the fundamental actions required for effective ranking. In this stage, the Reranker focuses on learning how to predict the correct intermediate actions $a_t$ based on the given task and context. To achieve this, the Reranker is trained on a dataset of expert demonstrations, denoted as $D_e$. This enables the model to acquire basic instruction-following capabilities and leverage prior knowledge. The training objective is to maximize the likelihood of the expert's document selection decisions:

$$\mathcal{J}_{BC}(\theta) = \mathbb{E}_{(q,D,k^*)\sim\mathcal{D}_e} \left[ \log \pi_\theta(k|q, D) \right], \quad (2)$$

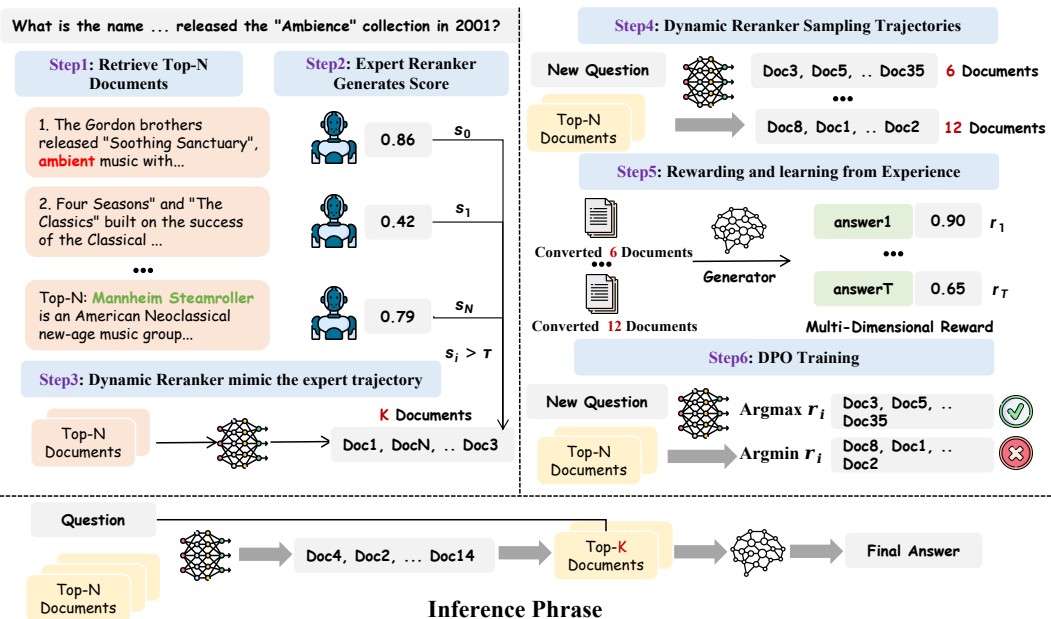

Figure 3: Illustration of our training framework. During the training phase, we have a total of six steps. First, we retrieve the Top-$N$ documents based on the given question. Then, we use an expert model to score each document and filter a subset of data for behavior cloning by the dynamic reranker. Next, we use the trained dynamic reranker to sample multiple different trajectories. The responses generated by the generator serve as rewards to evaluate the quality of the trajectories, and we select the trajectory pairs with the highest and lowest rewards as the training data for DPO. During the inference phase, DynamicRAG only require two LLM inferences.

where $q$ denotes the query, $D = \{d_1, d_2, ..., d_N\}$ represents the set of retrieved documents, $k^*$ is the expert-demonstrated optimal number of documents, and $\pi_\theta(k|q, D)$ represents the conditional probability of selecting $k$ documents from the candidate set $D$ given the query $q$.

### 2.2.2 Optimizing the Reranker via Exploration and Feedback

After the initial training with behavioral cloning, the Reranker requires further refinement to align its actions with the goal of maximizing response quality. This is achieved through an interactive learning process, in which the Reranker continually interacts with the environment to gather feedback, progressively improving its action policy.

The primary objective is to train a policy $\pi_\theta$ that maximizes the expected reward across all trajectories $\tau$ for a given environment $\mathcal{G}$ and user instruction $i$:

$$\mathcal{J}_{\text{explore}}(\theta) = \mathbb{E}_{\mathcal{G},i\sim\mathbb{G}}\mathbb{E}_{\tau\sim\pi_\theta(\tau|\mathcal{G},i)}[\mathbb{R}(\mathcal{G}, i, \tau)], \tag{3}$$

where $\mathbb{R}(\mathcal{G}, i, \tau)$ quantifies the response quality. However, optimizing this objective is challenging due to the complexity of the action space and the inherent inefficiencies in sampling. To address this, we adopt the Direct Preference Optimization (DPO) framework [44], which simplifies the reward optimization with pairwise comparisons of trajectories.

Formally, given a set of sampled trajectories $\{\tau_i\}_{i=1}^N$ with rewards $\{r_i\}_{i=1}^N$, we identify the trajectory pair $(\tau^+, \tau^-)$ such that:

$$\tau^+ = \arg\max_{\tau_i} r_i, \quad \tau^- = \arg\min_{\tau_i} r_i. \tag{4}$$

The whole training objective is then defined as:

$$\mathcal{J}_{\text{DPO}}(\theta) = \mathbb{E}_{(\tau^+,\tau^-)}\Big[\log\sigma\big(\beta(\log\pi_\theta(\tau^+) - \log\pi_\theta(\tau^-))\big)\Big], \tag{5}$$

$$\pi_\theta(\tau^+) = \frac{\pi_\theta(\tau^+|\mathcal{G}, i)}{\pi_{\text{ref}}(\tau^+|\mathcal{G}, i)}, \quad \pi_\theta(\tau^-) = \frac{\pi_\theta(\tau^-|\mathcal{G}, i)}{\pi_{\text{ref}}(\tau^-|\mathcal{G}, i)}. \tag{6}$$

This objective encourages the Reranker to assign higher probabilities to trajectories with greater rewards.

By iteratively combining environment feedback and RL optimization, the Reranker transitions from a basic policy to a robust, task-specific system capable of generating high-quality responses.

### 2.2.3 Reward Function Design

To evaluate the quality of the generated response $\hat{y}$ in relation to the ground-truth answer $y_{gt}$ and the contribution of reranked documents $\{D_i\}_{i=1}^K$, we employ a multi-dimensional reward function. This function integrates five key aspects of quality: Exact Match (EM), Semantic Similarity (SS), Textual Fluency (TF), Length Penalty (LP) and LLM-Based Evaluation (LLM-Eval). These dimensions collectively provide a holistic assessment of the generated output.

Formally, the reward function is defined as:

$$r(\mathcal{G}, i, \tau) = \alpha \cdot \text{EM} + \beta \cdot \text{SS} + \gamma \cdot \text{TF} + \lambda \cdot \text{LP} + \delta \cdot \text{LLM-Eval} \tag{7}$$

where EM, SS, TF, LP, and LLM-Eval are computed as functions of $(y_{gt}, \hat{y})$ and $\alpha$, $\beta$, $\gamma$, $\lambda$, and $\delta$ are weighting coefficients for each quality dimension.

The individual components are defined as follows:

- **Exact Match (EM):** Measures whether $\hat{y}$ matches $y_{gt}$ exactly:

$$\text{ExactMatch}(y_{gt}, \hat{y}) = \begin{cases} 1 & \text{if } \hat{y} = y_{gt}, \\ 0 & \text{otherwise.} \end{cases} \tag{8}$$

- **Semantic Similarity (SS):** Assesses the alignment between $\hat{y}$ and $y_{gt}$ using BERTScore [64]:

$$\text{SemanticSimilarity}(y_{gt}, \hat{y}) = \text{BERTScore}(y_{gt}, \hat{y}). \tag{9}$$

- **Textual Fluency (TF):** Evaluates fluency using ROUGE [28] metrics:

$$\text{TextualFluency}(y_{gt}, \hat{y}) = \text{ROUGE}(y_{gt}, \hat{y}). \tag{10}$$

- **Length Penalty (LP):** Encourages concise answers by penalizing longer responses:

$$\text{LengthPenalty}(\hat{y}) = \frac{1}{1 + \text{len}(\hat{y})}. \tag{11}$$

- **LLM-Based Evaluation (LLM-Eval):** Uses LLM-based scoring to assess alignment with task requirements, where $\mathcal{P}$ denotes the scoring prompt, detailed in Appendix D.3:

$$\text{LLM-Eval}(y_{gt}, \hat{y}) = \text{LLMScore}(\mathcal{P}, y_{gt}, \hat{y}). \tag{12}$$

## 3 Experiment

### 3.1 Datasets and Evaluation Metrics

We evaluate DynamicRAG's performance on comprehensive knowledge-intensive question-answering tasks, spanning over seven datasets and covering different types of challenges: Natural Questions [25], TriviaQA [19], HotpotQA [59], 2WikiMQA [15], FEVER [55], ASQA [51], and ELI5 [10]. For the first five datasets (NQ, TriviaQA, HotpotQA, 2WikiMQA, and ASQA), we follow prior studies [1, 62] and adopt exact match as the evaluation metric. For TriviaQA and FEVER, we used accuracy, while for ELI5, we employed ROUGE-L scores to assess performance [62, 41].

### 3.2 Baselines

**Baselines without Retrieval** We evaluate publicly available close-sourced LLMs, including GPT-3.5-turbo, GPT-4, and GPT-4o. These models represent state-of-the-art LLMs that are not augmented with external retrieval information. The system prompts and instruction formats are shown in Appendix D.3.

Table 1: The DynamicRAG results for different datasets among different backbone models. Results are directly from the original paper. Best results are in **bold** and the second results are underlined. * denotes that FLARE is based on the more powerful 175B text-davinci-003 model, which we currently do not have access to.

| Metrics | Extra Data for Training | NQ EM | TriviaQA EM/Acc | HotpotQA EM | 2WikimQA EM | ASQA EM | FEVER Acc | ELI5 Rg |
|---|---|---|---|---|---|---|---|---|
| *Baseline Without Retrieval* | | | | | | | | |
| GPT-3.5-Turbo [40] | N/A | 38.6 | 70.7/74.3 | 29.9 | 23.9 | 68.3 | 82.7 | 27.5 |
| GPT-4 [39] | N/A | 40.3 | 73.3/78.4 | 34.5 | 29.8 | 71.9 | 87.7 | 30.3 |
| GPT-4o [38] | N/A | 40.0 | 74.0/79.2 | 36.1 | 33.3 | 74.1 | 86.3 | 30.2 |
| *Baseline With Retrieval* | | | | | | | | |
| IRCoT [57] | N/A | - | - | 17.4 | - | - | - | - |
| ReAct [60] | N/A | - | - | 35.1 | - | - | 62.0 | - |
| RA-DIT [29] | ∼ 1,129k | 43.5 | 72.8/- | 36.6 | - | - | 86.9 | - |
| FLARE* [18] | N/A | - | - | - | 51.0 | 41.3 | - | - |
| Reward-RAG [36] | Unknown | 42.2 | 75.6/80.4 | - | - | - | 89.8 | - |
| LLaMA2-7B [56] | N/A | 17.9 | -/42.5 | 16.6 | 17.9 | 19.0 | 30.0 | 15.6 |
| w/ Reranker | N/A | 20.6 | -/49.6 | 18.9 | 18.3 | 21.1 | 35.6 | 16.7 |
| LLaMA2-7B-SFT | ∼ 130k | 29.1 | 53.7/59.1 | 27.1 | 18.9 | 23.8 | 40.6 | 18.6 |
| Self-RAG (LLaMA2-7B) [1] | ∼ 150k | 36.4 | -/66.4 | - | - | 30.0 | - | - |
| DRAGIN [52] | N/A | - | - | 23.2 | 22.0 | - | - | - |
| Smart-RAG (LLaMA2-7B) [12] | ∼ 218k | - | - | - | - | 26.6 | - | - |
| **Ours (LLaMA2-7B)** | ∼ 150k | **38.7** | **59.6/70.5** | **29.4** | **23.1** | **41.1** | **51.2** | **22.6** |
| LLaMA2-13B [56] | N/A | 23.6 | -/47.0 | 17.7 | 18.7 | 20.5 | 30.2 | 19.9 |
| w/ Reranker | N/A | 26.5 | -/53.2 | 20.4 | 18.8 | 23.6 | 37.1 | 20.3 |
| LLaMA2-13B-SFT | ∼ 130k | 32.5 | 60.1/66.2 | 27.9 | 19.1 | 28.4 | 45.8 | 20.1 |
| Self-RAG (LLaMA2-13B) [1] | ∼ 150k | - | -/69.3 | - | - | 31.7 | - | - |
| **Ours (LLaMA2-13B)** | ∼ 150k | **39.1** | **62.3/72.6** | **30.1** | **25.0** | **46.4** | **77.2** | **23.3** |
| LLaMA3-8B [13] | N/A | 36.4 | -/57.4 | 26.1 | 24.6 | 24.9 | 34.6 | 24.0 |
| w/ Reranker | N/A | 37.5 | -/64.5 | 28.7 | 25.3 | 29.8 | 49.7 | 23.7 |
| LLaMA3-8B-SFT | ∼ 130k | 39.1 | 67.5/74.2 | 31.5 | 27.1 | 46.8 | 82.1 | 22.9 |
| Auto-RAG (LLaMA3-8B-Instruct) [61] | Unknown | 37.9 | 60.9/- | - | - | 30.0 | - | - |
| ChatQA-1.5 (LLaMA3-8B) [32] | ∼ 442k | 42.4 | 81.0/87.6 | 33.4 | 26.8 | - | 90.9 | - |
| RankRAG (LLaMA3-8B) [62] | ∼ 470k | **50.6** | **82.9/89.5** | 35.3 | 31.4 | - | **93.8** | - |
| **Ours (LLaMA3-8B)** | ∼ 150k | 48.4 | 78.3/87.4 | **36.7** | **34.2** | **56.3** | 91.4 | **24.6** |

**Baselines with Retrieval**    We compare our method against several retrieval-augmented baselines. The baselines are categorized into four groups as follows: **RAG-based Baselines**: This group includes approaches such as IRCoT [57], ReAct [60], FLARE [18], RA-DIT [29] and Reward-RAG [36], which leverage agent-like strategies for retrieval-augmented generation. **LLaMA2-7B-based Baselines**: This category consists of standard retrieval-augmented baselines such as Vanilla-RAG, as well as its enhanced versions with additional components like reranker, SFT, Self-RAG [1], DRAGIN [52] and Smart-RAG [12]. **LLaMA2-13B-based Baselines**: Similar to the LLaMA2-7B group, this set includes Vanilla-RAG, Reranker, SFT, and Self-RAG [1], providing a larger-scale comparison using the LLaMA2-13B model. **LLaMA3-8B-based Baselines**: In this group, we consider models based on LLaMA3-8B, including Vanilla-RAG and its variations with Reranker and SFT. Additionally, we compare our models with more advanced retrieval-augmented methods such as Auto-RAG [61], ChatQA-1.5 [32], and RankRAG [62].

### 3.3   Implementation Details

**Training data and settings**    Our training data comprises 150k diverse instruction-output pairs, drawn from Alpaca [54], KILT [41], ASQA [51], and OpenBookQA [35]. We employ three models as the base LMs for our dynamic reranker and generator: LLaMA2-7B, LLaMA2-13B, and LLaMA3-8B. For the retriever model, we use the off-the-shelf Contriever-MS MARCO [16] as the default retriever, retrieving up to 45 documents for LLaMA3 and 20 documents for LLaMA2 per input, tailored to their respective context window sizes. Unless otherwise specified, the retriever and generator share the parameters. Additional training details can be found in the Appendix D.

**Inference settings**    For the dynamic reranker, we set the temperature to 0.2 to enhance output diversity. For the generator, we use a temperature of 0 to ensure output stability and reproducibility. By default, we use the top 45 documents from Contriever-MS MARCO [16] as input to the reranker. In contrast, all baseline methods use the top 10 documents from Contriever-MS MARCO as input to ensure a fair comparison.

Table 2: The performance of different Reranker models. Results are directly from the original paper. Best results are in **bold** and the second results are underlined.

| Metric | Training Data | NQ | | | HotpotQA | | | Avg. |
|---|---|---|---|---|---|---|---|---|
| | | R@5 | R@10 | R@20 | R@5 | R@10 | R@20 | |
| *Close-Source Models* | | | | | | | | |
| GPT-3.5-Turbo [40] | Unknown | 77.8 | 82.5 | 85.7 | 52.1 | 56.6 | 62.4 | 69.5 |
| GPT-4 [39] | Unknown | 79.3 | 83.2 | 85.1 | 53.2 | 57.0 | 61.0 | 69.8 |
| *Open-Source Rerank Models* | | | | | | | | |
| BM25 [46] | N/A | 38.0 | 50.7 | 60.1 | 57.5 | 63.0 | **67.5** | 56.1 |
| Contriever [16] | Unknown | 73.6 | 80.2 | 84.8 | 53.1 | 58.7 | 62.4 | 68.8 |
| monoT5 [37] | $\sim$ 503k | 75.6 | 80.9 | 84.9 | 54.8 | 60.2 | 63.3 | 70.0 |
| RankLLaMA [34] | $\sim$ 503k | 77.8 | 83.1 | 86.0 | 57.1 | 62.1 | 64.8 | 71.8 |
| ChatQA-1.5 (LLaMA3-8B) [32] | N/A | 68.2 | 75.7 | 82.0 | 37.4 | 45.0 | 53.6 | 60.3 |
| RankRAG (LLaMA3-8B) [62] | $\sim$ 50k | **80.3** | **84.0** | 86.3 | 57.6 | 61.8 | 65.2 | 72.5 |
| *Open-Source Generative Models* | | | | | | | | |
| GENRE [4] | $\sim$ 406k | 61.4 | - | - | 34.0 | - | - | - |
| Re3eval [50] | $\sim$ 240k | 65.4 | - | - | 44.2 | - | - | - |
| SEAL [2] | Unknown | 68.2 | - | - | 51.0 | - | - | - |
| **DynamicRAG (LLaMA3-8B)** | $\sim$ 20k | 79.3 | 83.7 | **86.8** | **59.1** | **63.7** | 67.2 | **73.7** |

## 3.4 Main Results

### 3.4.1 Comparison against baselines with and without retrieval

Table 1 presents a comprehensive comparison of our proposed DynamicRAG approach against various baseline models, categorized into those without retrieval and those incorporating retrieval mechanisms. For baseline models that do not utilize the retrieval, such as GPT-3.5-Turbo, GPT-4, and GPT-4o, our approach significantly outperforms them across multiple datasets. Notably, in the NQ dataset, our method achieves an EM score of 48.4 with LLaMA3-8B, surpassing GPT-4o. These results highlight the effectiveness of our retrieval-augmented approach compared to models without retrieval capabilities. When compared to agent-based baselines such as IRCoT, ReAct, and Reward-RAG, our DynamicRAG framework consistently achieves state-of-the-art performance across various datasets. Moreover, our approach achieves superior performance compared to other retrieval-based models such as RankRAG and ChatQA-1.5, despite using significantly less training data ($\sim$150k examples vs. $\sim$470k for RankRAG). Second, the effectiveness of our method is consistent across different backbone sizes (7B, 13B, 8B), showing scalability and robustness.

### 3.4.2 Comparison with baselines for reranking performance

The Table 2 presents a comparison of different reranker models categorized into three groups: close-sourced models, open-sourced rerank models, and open-sourced generative models. Our LLaMA3-8B-based model, DynamicRAG, demonstrates competitive performance while utilizing only 20k training samples, achieving results comparable to RankRAG, which requires 50k training samples. Furthermore, our model significantly surpasses other open-sourced models, such as monoT5, RankLLaMA, and generative models like Re3eval, across key ranking metrics (R@5, R@10, and R@20). This highlights the effectiveness of our approach in efficiently utilizing limited training data without compromising performance.

Additionally, our approach adopts a list-wise reranking strategy, which contributes to superior overall ranking efficiency compared to other models that primarily rely on point-wise methods. Notably, we leverage the quality of generated responses as a signal for reranking, which significantly enhances model performance, particularly when compared to traditional information retrieval-based models.

## 3.5 Ablation Studies

We perform various ablation studies to understand the importance of different factors in DynamicRAG.

### 3.5.1 Impact of Reranker and Generator Size

The Table 3a presents results from different Reranker and Generator configurations in the DynamicRAG, evaluating model variants with LLaMA2-7B, LLaMA2-13B, and LLaMA3-8B, where

(a) Results of different models and sizes for Reranker and Generator. * denotes we share the parameters of the Reranker and Generator. We use Exact Match as the metric.

| DynamicRAG | | NQ | HotpotQA | Avg. |
|---|---|---|---|---|
| Reranker | Generator | EM | EM | |
| LLaMA2-7B | LLaMA2-13B | 37.6 | 28.6 | 33.1 |
| LLaMA2-13B | LLaMA2-13B | 38.7 | 29.8 | 34.3 |
| LLaMA2-13B* | LLaMA2-13B* | 39.1 | 30.1 | 34.6 |
| LLaMA3-8B | LLaMA2-13B | 39.4 | 30.2 | 34.8 |

(b) The impact of different key components in DynamicRAG among different benchmarks. We use Exact Match as the metric.

| | NQ | HotpotQA | ASQA | Avg. |
|---|---|---|---|---|
| | EM | EM | EM | |
| **DynamicRAG** | 48.4 | 36.7 | 56.3 | 47.1 |
| w/o Retrieval | 25.0 | 25.6 | 15.7 | 22.1 |
| w/o Reranking | 36.4 | 27.2 | 39.8 | 34.5 |
| w/o RL | 44.6 | 29.6 | 45.5 | 39.9 |

performance improves as model size increases. The 13B Reranker and 13B Generator configuration outperforms the 7B-13B setup in both NQ and HotpotQA, with the average EM score rising from 33.1 to 34.3. Switching to an 8B Reranker with a 13B Generator results in a slight further increase in the average EM score to 34.8, suggesting that a larger Reranker can enhance performance, even with a fixed Generator size. The model where both the Reranker and Generator share parameters (denoted by *) achieves the EM of 39.1 on NQ and 30.1 on HotpotQA, yielding an average EM score of 34.6. This improvement indicates that sharing parameters allows the Reranker and Generator to better complement each other, as their tasks can mutually enhance performance.

### 3.5.2 Effect of Key Components on DynamicRAG Performance

The Table 3b shows the performance of DynamicRAG under various ablation conditions, where key components such as retrieval, reranking, reinforcement learning, and iterative training are removed. Evaluation on NQ, HotpotQA, and ASQA reveals that removing retrieval causes a significant drop in performance, with EM scores falling to 25.0 on NQ, 25.6 on HotpotQA, and 15.7 on ASQA, leading to an average EM of just 22.1. This emphasizes the critical importance of retrieval in supplying relevant information. Excluding reranking results in a degradation in performance, especially on NQ (EM = 36.4), indicating that reranking has a beneficial effect. Removing RL also hinders performance across all datasets, with the average EM decreased to 39.9, particularly on NQ and ASQA (44.6, 45.5), suggesting that RL significantly benefits performance.

## 3.6 Model Analysis

### 3.6.1 Efficiency

**From LLM-Calling Perspective** We evaluate the maximum number of LLM calls required by different RAG models to generate an answer when the retriever returns 20 documents. It is evident that the closer a model is to the top-left corner, the better it performs, as both effectiveness and efficiency are optimized. Our model is positioned in the top-left corner, demonstrating superior performance compared to other models. Specifically, our model can generate answers with only two LLM calls when the retriever returns 20 documents.

**From Token Perspective** We empirically evaluated computational efficiency against existing methodologies, notably RankRAG. Our architecture processes Question + Top-20 documents concurrently, producing Top-k documents through a single Reranker pass before generating the final answer via Question + Top-k document integration. In contrast, RankRAG's methodology necessitates separate processing of Question + individual document pairs, requiring 20 distinct forward passes for a corpus of 20 documents. As-

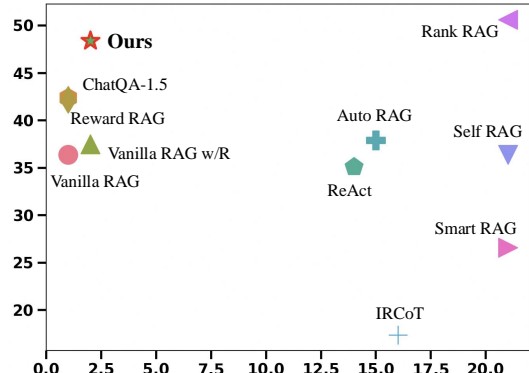

Figure 4: Comparison of different RAG models in terms of efficiency and effectiveness. The x-axis represents the number of LLM calls, while the y-axis denotes the average performance on the NQ benchmark. Models closer to the top-left corner achieve better overall performance.

suming $k = 10$ for both approaches, Table 4 presents mean runtime metrics (averaged across three experimental iterations):

Table 4: Comparative analysis of processing latency from token-input perspective.

| Input | Time (Seconds) |
| --- | --- |
| Question + Top-10 Docs (Vanilla-RAG) | 0.57 |
| Question + Top-20 Docs (4k context window) to rank | 0.75 |
| (Question + Single Doc) $\times$ 20 for reranking | 13.00 |
| Question + Top-k Docs (avg $k = 12$) | 0.61 |

Results demonstrate that contextual integration yields substantially higher computational efficiency compared to multiple LLM invocations. (As the RankRAG implementation is not publicly available, we constructed a functional equivalent utilizing LLaMA2-7B—matching the original architecture's scale—deployed within a VLLM framework on 8 A100 GPUs.) Our methodology demonstrates approximately $17\times$ superior throughput relative to RankRAG's sequential scoring approach. Furthermore, compared to standard RAG pipelines without reranking, our approach introduces minimal computational overhead—specifically, a $2.3\times$ latency increase while delivering significant performance gains. For instance, on the NQ benchmark, our methodology demonstrates a 9.6 percentage point improvement over vanilla RAG implementations using LLaMA2-7B.

### 3.6.2 Reranked Document Distribution

We analyzed the reranked results of DynamicRAG on NQ and HotpotQA, as shown in Figure 5 [2]. Before RL training, the reranked results on NQ and HotpotQA predominantly had k values of 14 and 15. This is because the model trained solely with SFT tends to favor a higher number of reranked documents, achieving a better downstream performance. However, after RL training, especially with the introduction of the length penalty, a leftward shift in $k$ values can be observed, with peaks appearing at 12, 13, and 14. This indicates a tendency to output fewer reranked documents. This also proves the effectiveness of our RL training.

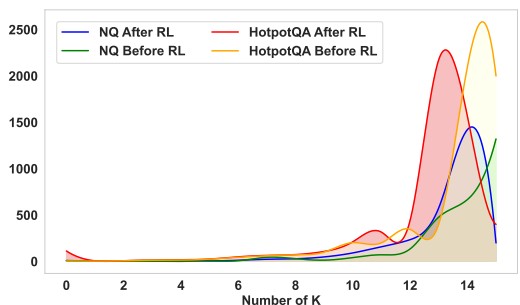

Figure 5: Distribution of reranked document numbers (k) on NQ and HotpotQA before and after RL training. $k$ is truncated at 15 to ensure a fair comparison, as we restrict $k \le 15$ during both training and sampling.

Due to space limitations, we defer several important analyses to the Appendices. Appendix C includes ablation studies on different retrievers, Top-$N$ document selection, reward function components, and training experiments with closed-source models. Appendix D provides comprehensive implementation details, qualitative examples demonstrating our dynamic reranking approach, and complete prompt templates.

## 4  Conclusion

This work introduces DynamicRAG, a new reinforcement learning framework to optimize reranking in RAG. By modeling the reranker as an RL agent and using LLM response quality as rewards, it dynamically adjusts the order and number of retrieved documents per query. This dynamic reranking mechanism enhances both the relevance of selected documents and the overall system efficiency. Extensive evaluations on seven knowledge-intensive datasets demonstrate that DynamicRAG consistently outperforms existing fine-tuned and prompting-based approaches, achieving state-of-the-art performance.

---

[2] We selected a maximum of 15 documents for reranking to maintain fairness in comparison with existing studies, which commonly evaluate top-10 documents. Given LLaMA-7B's 4K token context length and an average document length of approximately 200 tokens, accommodating 15 documents and their associated queries is comfortably feasible on an 80GB A100 GPU.

# 5    Acknowledgement

Research was supported in part by National Science Foundation IIS-19-56151, NSF IIS 25-37827, the Molecule Maker Lab Institute: An AI Research Institutes program supported by NSF under Award No. 2019897, and the Institute for Geospatial Understanding through an Integrative Discovery Environment (I-GUIDE) by NSF under Award No. 2118329. The research has used the Delta/DeltaAI advanced computing and data resource, supported in part by the University of Illinois Urbana-Champaign and through allocation #250851 from the Advanced Cyberinfrastructure Coordination Ecosystem: Services & Support (ACCESS) program, which is supported by National Science Foundation grants OAC 2320345, #2138259, #2138286, #2138307, #2137603, and #2138296. Any opinions, findings, and conclusions or recommendations expressed herein are those of the authors and do not necessarily represent the views, either expressed or implied, of DARPA or the U.S. Government.

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

# A Appendix

## A.1 Limitations

While DynamicRAG demonstrates strong performance across multiple benchmarks, several limitations warrant discussion.

- Our current work exclusively employs list-wise ranking methodologies, where the reranker processes all documents collectively. Point-wise and pair-wise ranking methods, which may offer different computational and quality trade-offs, remain unexplored and will be addressed in future research.

- our optimization approach treats document selection holistically without considering fine-grained step-level decisions. Exploring step-wise optimization, such as progressive refinement or adaptive termination criteria during the reranking process, remains as future work and could potentially lead to more efficient document processing and better interpretability.

- We adopt DPO [44] as the sole reinforcement learning algorithm in our main experiments due to its training stability and efficiency. While DPO's off-policy nature provides practical advantages, our framework's well-defined reward function makes it compatible with various on-policy algorithms such as PPO [48] and GRPO [49], which we leave for future exploration.

## A.2 Related Work

### A.2.1 Retrieval-Augmented Generation

RAG boosts LLM performance by adding external knowledge, enhancing factual accuracy and context [26, 14]. Research includes retrieval-based next-token prediction [22, 45] and end-to-end fine-tuning for better integration [3, 17, 65]. Asai et al. [1] optimizes RAG using special tokens for adaptive retrieval and reflection, fine-tuning with a critic model. In related work, Ke et al. [21] optimized RAG by training a bridge model to refine the retriever-LLM connection. While sharing similarities, our approach differs in key ways: (1) We optimize the reranker by treating it as an agent that interacts with the generator to collect training trajectories. (2) We jointly train the reranker and generator, finding that shared parameters improve adaptation to downstream tasks.

### A.2.2 LLM for Reranking

LLMs are increasingly used for passage reranking, with methods generally being point-wise (assessing individual relevance via relevance or query generation [27, 9, 47]), pair-wise (comparing passage pairs for relative relevance [43]), and list-wise (holistically ranking passages like Learning to Rank [53, 33, 5]). Recent advancements like zero-shot reranking with fine-tuned open-weight LLMs [42, 30] and logit-based methods [11, 7] aim to solve issues but often need specific fine-tuning or have scalability limits. Our approach treats the LLM reranker as an agent, initially using behavior cloning to imitate expert behavior, then interacting with the generator to create trajectories for further optimization.

## A.3 Algorithm

The algorithm of our main method is shown in Algorithm 1.

# B Preliminaries

## B.1 Retrieval Augmentation Generation

### B.1.1 Retrieval Phase

The first step in the RAG framework is to retrieve relevant documents or passages from a large corpus. This is typically done using an information retrieval (IR) system. Given a query $\mathbf{q}$, the IR system selects a set of relevant documents $D_1, D_2, \ldots, D_k$ based on some retrieval method, such as BM25, dense retrieval with embeddings, or a hybrid approach. Formally, the retrieval process can be

---

**Algorithm 1** DynamicRAG

---

**Require:** Expert dataset $\mathcal{D}_e$, environment $\mathcal{G}$, iterations $N$, Normal dataset $\mathcal{D}_{train}$
 1: INITIALIZE reranker $\pi_{\theta_r}$ and generator $\pi_{\theta_g}$
 2: **STEP 1: BEHAVIORAL CLONING**
 3: **for** each sample $(\mathcal{G}, i, \tau) \in \mathcal{D}_e$ **do**
 4:    Update reranker via:

$$\mathcal{J}_{BC}(\theta_r) = \mathbb{E}\Big[ \sum_{t=1}^{T} \log \pi_{\theta_r}(a_t | \mathcal{G}, i, h_{t-1}) \Big]$$

 5: **end for**
 6: **STEP 2: GENERATOR TRAINING**
 7: **for** each sample $(\mathbf{q}, \hat{D}, y_{gt}) \in \mathcal{D}_{train}$ **do**
 8:    Optimize generator $\pi_{\theta_g}$ via:

$$\mathcal{J}_{\text{gen}}(\theta_g) = \mathbb{E}\Big[ \sum_{t=1}^{T} \log p_{\theta_g}(y_t \mid \mathbf{q}, \hat{D}, y_{<t}) \Big]$$

 9: **end for**
10: **STEP 3: INTERACTIVE LEARNING**
11: **for** $n = 1$ to $N$ **do**
12:    Collect trajectories and compute rewards:

$$r = \alpha \cdot \text{EM} + \beta \cdot \text{SS} + \gamma \cdot \text{TF} + \lambda \cdot \text{LP} + \delta \cdot \text{LLM-Eval}$$

13:    Optimize reranker via DPO:

$$\mathcal{J}_{\text{DPO}}(\theta_r) = \mathbb{E}\Big[ \log \sigma\big(\beta(\log \pi_{\theta_r}(\tau^+) - \log \pi_{\theta_r}(\tau^-))\big) \Big]$$

14: **end for**

---

represented as:

$$D = \texttt{Retriever}(\mathbf{q}, \mathcal{C}), \tag{13}$$

where $\mathcal{C}$ is the document corpus, and $D$ represents the set of retrieved documents.

To quantify the relevance score $s(D_i, \mathbf{q})$ of each document $D_i$ with respect to the query $\mathbf{q}$, we define:

$$s(D_i, \mathbf{q}) = \text{Score}(D_i, \mathbf{q}), \tag{14}$$

where $\text{Score}(\cdot)$ is a function specific to the retrieval method employed (e.g., BM25 score, cosine similarity for dense embeddings).

### B.1.2 Encoding Phase

Once the relevant documents have been retrieved, both the query $\mathbf{q}$ and the documents $\{D_1, D_2, \ldots, D_k\}$ are encoded into fixed-size vectors using a neural encoder such as BERT [8], RoBERTa [31], or other transformer-based models. The goal is to obtain a dense representation of both the query and documents. Let $\mathbf{q}_{\text{enc}}$ and $\mathbf{D}_i$ represent the encoded query and the encoding of document $D_i$, respectively:

$$\begin{aligned} \mathbf{q}_{\text{enc}} &= \text{Encode}(\mathbf{q}), \\ \mathbf{D}_i &= \text{Encode}(D_i) \quad \text{for} \quad i = 1, 2, \ldots, k. \end{aligned} \tag{15}$$

### B.1.3 Generation Phase

After encoding, the next phase is to generate an answer or response based on the query and retrieved documents. The generation model takes both the query and the retrieved documents as input and generates a response $\hat{y}$. The generation process can be framed as maximizing the conditional probability:

$$\hat{y} = \text{argmax}_y\, p(y \mid \mathbf{q}, D_1, D_2, \ldots, D_k), \tag{16}$$

where $p(y \mid \mathbf{q}, D_1, D_2, \ldots, D_k)$ is the likelihood of generating the output $y$ given the query $\mathbf{q}$ and the retrieved documents. To incorporate the encoded representations, we model the conditional probability as:

$$p(y \mid \mathbf{q}, D_1, \ldots, D_k) = \text{Decode}(\mathbf{q}, D_1, \ldots, D_k). \tag{17}$$

Finally, the generation model is trained to maximize the likelihood of generating the correct response given the query and the retrieved documents. The loss function can be expressed as:

$$\mathcal{L} = -\log p(y \mid \mathbf{q}, D_1, D_2, \ldots, D_k), \tag{18}$$

where $y$ is the ground-truth response and $\mathbf{q}, D_1, \ldots, D_k$ are the input query and retrieved documents.

### B.2 Learning from Environment

We model the Reranker as a player and the generator as an environment $\mathcal{G}$. We can formalize the reranker's task in the environment as a partially observable Markov decision process $(\mathcal{I}, \mathcal{S}, \mathcal{A}, \mathcal{T}, \mathbb{R})$, where $\mathcal{I}$ is the instruction space, $\mathcal{S}$ is the state space, $\mathcal{A}$ is the action space, $\mathcal{T} : \mathcal{S} \times \mathcal{A} \to \mathcal{S}$ is the deterministic state transition function, and $\mathbb{R} : \mathcal{S} \times \mathcal{A} \to \mathcal{R}$ is the reward function. In our design, we exclude explicit observations $\mathcal{O}$ here since we focus on the overall reward of the reranker's complete episode in the environment, rather than step-wise rewards. We leave observation-based optimization for future work.

Given a task instruction $i$ in environment $\mathcal{G}$, the Reranker generates an action sequence $a_1, a_2, \ldots, a_T$ based on its policy $\pi_\theta$, where each action $a_t \sim \pi_\theta(\cdot | \mathcal{G}, i, a_1, \ldots, a_{t-1})$ is determined by the history of previous actions. The trajectory is represented as:

$$\tau = (a_1, \ldots, a_T) \sim \pi_\theta(\tau | \mathcal{G}, i) \tag{19}$$

$$\pi_\theta(\tau | \mathcal{G}, i) = \prod_{t=1}^{T} \pi_\theta(a_t | \mathcal{G}, i, h_{t-1}) \tag{20}$$

where $T$ is the number of steps in interaction, and $h_{t-1} = (a_1, \ldots, a_{t-1})$ represents the history of action sequences up to step $t-1$. The final reward $r(\mathcal{G}, i, \tau) \in [0, 1]$ is computed based on the quality of the generator's response.

## C Further Analysis

### C.1 Performance with different retrievers

We conducted experiments to compare the performance of Vanilla-RAG and DynamicRAG on three different benchmarks: NQ, HotpotQA, and ASQA, using different retrievers (DPR [20], Contriever, MonoT5), as shown in Figure 6. It can be observed that as the retriever models improve, both approaches exhibit better performance on downstream tasks. This also demonstrates the robustness of our model.

### C.2 Effect of Top-N Documents on DynamicRAG Performance

The Figure 7 presents the performance of DynamicRAG with different numbers of top-K documents (from 50 to 500) used for reranking across three benchmarks: NQ, HotpotQA, and ASQA. We adopt

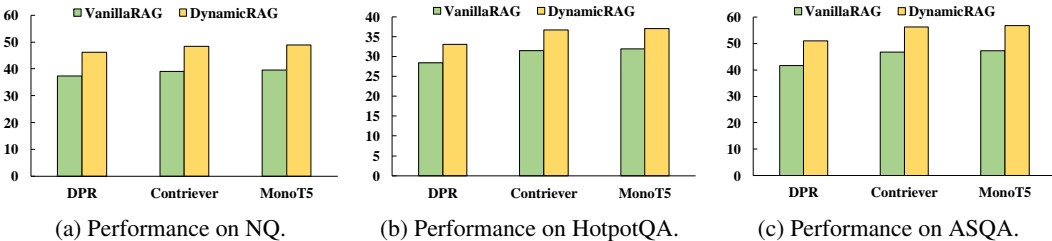

(a) Performance on NQ.     (b) Performance on HotpotQA.     (c) Performance on ASQA.

Figure 6: Performance with different retrievers between Vanilla-RAG and DynamicRAG.

the same technique as Sun et al. [53], where we use sliding window to handle large document sets which allows the model to process larger corpora efficiently. First, as the number of top documents increases, the performance improves, reaching its peak around Top-100 or Top-150, with the highest EM scores recorded for NQ and HotpotQA. However, as more documents are introduced (Top-200, Top-300, Top-500), the performance starts to decline slightly, with the average EM dropping from 56.8 (Top-50) to 55.3 (Top-500). This trend indicates that beyond a certain threshold, increasing the number of documents introduces irrelevant or misleading information that negatively impacts the model's performance. The challenge of processing and filtering through a larger set of documents, which can introduce noise and reduce the model's ability to focus on the most relevant content.

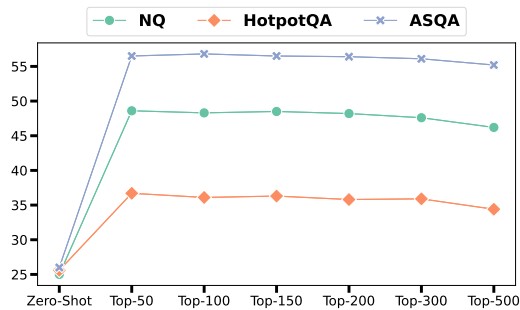

Figure 7: The impact of varying the number of Top-$N$ documents (Top-50, Top-100, Top-150, Top-200, Top-300, and Top-500) used for reranking on DynamicRAG performance across different benchmarks (NQ, HotpotQA, ASQA). We use Exact Match as the metric.

### C.3   Impact of Different Reward Functions on DynamicRAG Performance

The Table 5 presents an analysis of DynamicRAG's performance under various reward model configurations, incorporating key components such as EM (Exact Match), SS (Semantic Similarity), TF (Textual Fluency), LP (Length Penalty), and LLMEval (LLM-Based Evaluation). The EM component is found to be critical for open-domain QA tasks, as its removal results in a significant performance decline, particularly on benchmarks such as NQ, HotpotQA, and ASQA. However, its impact on long-text generation tasks, like ELI5, is comparatively less pronounced. Conversely, the removal of SS or TF functions yields opposing effects: removing SS and TF has a more pronounced negative impact on long-text generation tasks, while their effect on open-domain QA is relatively modest. This suggests that SS is essential for improving the model's generalization capabilities, while TF plays a crucial role in enhancing text relevance. The LP function, when excluded, results in a slight but consistent drop in performance across all benchmarks, indicating its influence on overall model balance by regulating output length and maintaining response coherence. LLMEval, exhibiting effects similar to those of SS and TF, contributes moderately to performance optimization, underscoring its supportive role. Overall, the consistent trends across different reward function configurations highlight that the model's success is predominantly driven by the synergy of EM, SS, TF, and LP, with LLMEval serving as a supplementary component in refining performance.

Table 5: The performance of DynamicRAG with different reward function designs across various benchmarks (NQ, HotpotQA, ASQA, ELI5). We use Exact Match as the metric for NQ, HotpotQA and ASQA and use Rouge-L as the metric for ELI5.

| Reward Function | NQ EM | HotpotQA EM | ASQA EM | ELI5 Rg | Avg. |
|---|---|---|---|---|---|
| **DynamicRAG** | 48.4 | 36.7 | 56.3 | 24.6 | 41.5 |
| w/o EM | 39.6 | 22.3 | 35.4 | 24.4 | 30.4 |
| w/o SS | 47.7 | 36.0 | 55.6 | 22.6 | 40.5 |
| w/o TF | 47.8 | 36.2 | 55.7 | 22.0 | 40.4 |
| w/o LP | 48.0 | 36.2 | 55.8 | 24.0 | 41.0 |
| w/o LLMEval | 48.1 | 36.3 | 56.0 | 22.7 | 40.8 |

To further prove the robustness of our reward functions, we additionally experimented with the following configurations ($\alpha$, $\beta$, $\gamma$, $\lambda$, and $\delta$):

- Hyperparameter 1 = (0.3, 0.25, 0.2, 0.15, 0.1)
- Hyperparameter 1 = (0.4, 0.15, 0.15, 0.15, 0.15)
- Hyperparameter 1 = (0.1, 0.25, 0.2, 0.15, 0.3)

Table 6: Performance comparison across different settings and datasets

| Setting | NQ | HotpotQA | ASQA | ELI5 | Avg. |
|---|---|---|---|---|---|
| Paper Setting | 48.4 | 36.7 | 56.3 | 24.6 | 41.5 |
| Hyperparameter 1 | 48.5 | 36.6 | 56.2 | 24.3 | 41.4 |
| Hyperparameter 2 | 48.5 | 36.7 | 56.5 | 24.4 | 41.5 |
| Hyperparameter 3 | 48.2 | 36.6 | 56.2 | 24.4 | 41.3 |

The results show that different parameter settings yield similar performance, indicating that our framework is robust and largely insensitive to these hyperparameters.

## C.4 Dynamic Reranker Module Enhances Closed-Source Model RAG Performance

Many studies [62, 6] have demonstrated that increasing the number of input documents introduces more irrelevant information, ultimately degrading model performance. Therefore, dynamically adjusting $k$ remains crucial, even for strong models. In fact, since our reward mechanism leverages responses and ground-truth outputs, our approach is fully applicable to closed-source, robust models such as GPT-4o. Experimental results validating this claim are presented below:

Table 7: Performance Improvement Using Dynamic Reranker on Closed-Source Models.

| Model | NQ | HotpotQA | ASQA |
|---|---|---|---|
| GPT-4o | 40.0 | 36.1 | 74.1 |
| GPT-4o w/ RAG Top-20 | 40.4 | 34.2 | 73.2 |
| GPT-4o w/ Dynamic Reranker | 42.3 | 36.9 | 74.8 |

The results show that the Dynamic Reranker module consistently improves GPT-4o's performance across all datasets, with gains of 2.3, 0.8, and 0.7 percentage points on NQ, HotpotQA, and ASQA, respectively. In contrast, the standard RAG Top-20 approach shows inconsistent results, even degrading performance on two datasets. The Dynamic Reranker module effectively addresses this issue by dynamically adjusting the number of reranked documents k, demonstrating that even powerful closed-source models like GPT-4o can benefit from our approach.

## C.5 Training Data Scaling for DynamicRAG

Our main experiments were conducted using a training dataset of 150k samples. To investigate the impact of training data scale, we expanded the dataset incrementally to 160k, 180k, and 200k samples from the same training corpus, primarily augmenting the training data used in the RL phase. The results of this scaling analysis are presented in Table 8.

Table 8: Performance comparison with increasing training data size

| Model | Training Data | NQ | HotpotQA | ASQA | ELI5 | Avg. |
|---|---|---|---|---|---|---|
| | 150k | 48.4 | 36.7 | 56.3 | 24.6 | 41.5 |
| | 160k | 48.6 | 36.8 | 56.6 | 24.6 | 41.7 |
| DynamicRAG-8B | 180k | 48.7 | 37.0 | 57.0 | 24.8 | 41.8 |
| | 200k | 49.0 | 37.0 | 57.2 | 24.9 | 42.0 |

As demonstrated by the results, the performance of our model consistently improves as the training data size increases. This trend is particularly noteworthy given that in our main experiments, DynamicRAG did not outperform RankRAG on three of the benchmarks. However, it is important to highlight that our model was trained with less than one-third of the training data used for RankRAG. The steady improvement in performance with increased data volume suggests that with comparable training data sizes, DynamicRAG has the potential to match or exceed the performance of RankRAG.

Table 9: Full list of instructions used during our evaluations. We use the same prompt when eval Open-domain QA (NQ, TriviaQA, HotpotQA, 2WikimQA, ASQA.)

| Dataset | Instruction |
|---|---|
| ARC | Please answer the following questions and directly output the answer options. |
| FEVER | Please answer the question with "SUPPORTS", "REFUTES" or "NEI" based on what you know. |
| ELI5 | Please answer the question with a paragraph. |
| Open-domain QA | Please answer the question with a short phrase. |

# D   Experiment Details

## D.1   Training Details

We train our models on 8 Nvidia A100 GPUs, each with 80GB of memory. Fine-tuning is performed for 1 epoch with an effective batch size of 4, achieved by setting a per-device batch size of 2 and accumulating gradients over 2 steps. The learning rate is configured at 1e-5, with a 10% warmup ratio and a cosine learning rate schedule. Maximum token lengths are set to 4,096 for LLaMA2 and 8,192 for LLaMA3, adhering to the training configuration. Multi-GPU distributed training is efficiently handled using DeepSpeed Stage 3, with Bfloat16 precision to optimize memory usage. FlashAttention enhances the efficiency of long-context training. For behavior cloning, we employ MonoT5 [37] as the expert model, set $\tau$ as 0.8 and constrain the number of documents to a maximum of 15, since many works [1, 62] only use the top-10 as the input for the generator, we aim to obtain a relatively fair comparison, so we do not set $k$ too large. For $\alpha$, $\beta$, $\gamma$, $\lambda$, and $\delta$ in reward functions, we simply use (0.2, 0.2, 0.2, 0.2, 0.2). For sampling trajectories, we use temperature to 1.0 and top p to 0.9. For DPO training, we used the following configuration: a learning rate of 5e-6, 2 epochs, a 10% warmup ratio, and a cosine learning rate schedule. The batch size was set to 1, and accumulating gradients over 4 steps. For inference, we leverage the same A100 GPUs and utilize vLLMs to accelerate inference time.

### D.1.1   Retriever Setting

To construct the training data, we retrieved the top 45 documents using Contriever-MS MARCO from official 2018 English Wikipedia embeddings. These documents were used to create training datasets for both the reranker and generator components. Unlike other works, we do not use retrieval results from external sources, such as Google Search. Instead, all evaluations are strictly based on retrieval results from the same retriever used for training data construction, ensuring consistency and comparability.

### D.1.2   Training Dataset

We utilized the Alpaca and specific KILT benchmark datasets, including NQ, FEVER, HotpotQA, ELI5, and TriviaQA, as well as ASQA and OpenBookQA, totaling approximately 150k data instances. And we employ 20k for cold-start reranker training, 100k for supervised fine-tuning of the generator, and 30k for DPO training.

### D.1.3   Evaluation Setting

For ELI5, we set the maximum token length to 256 to accommodate the benchmark's requirement for long-answer responses. For all other benchmarks, the maximum token length is set to 50. Table 9 shows the list of the instructions used during evaluations.

## D.2   Qualitative Examples

Tables 10 and 11 present two distinct examples illustrating the effectiveness of our approach. In the first example, the dynamic reranker produces a reordered sequence and selects a different number of retrieved documents compared to the expert model, yet it successfully generates the correct answer. This demonstrates the reranker's superior ability to identify the most relevant information. In the second example, no additional documents are selected. This suggests that the reranker recognizes the

query as sufficiently straightforward for the generator, eliminating the need for external information and thereby enhancing the overall efficiency of the generation process.

## D.3   DynamicRAG Prompt

DynamicRAG includes three prompts: the prompt used for constructing dynamic reranker data, the prompt used for constructing retrieval-based generator data, the prompt for the reward function, and the prompt for GPT and LLaMA baselines as shown in Tables 12, 13, 14, 15, and 16 respectively.

Table 10: Case Study for DynamicRAG. We compare our method with Vanilla-RAG with Reranker (monoT5). The initial retrieved content is the same for both methods. We take Top-40 documents as the input.

| Question | What is the name of the American Neoclassical new-age music group that released the "Ambience" collection in 2001? |
| --- | --- |
| GPT-4o | **The Ambient.** ✗ |
| Vanilla-RAG w/Reranker | **(Reranker Results:)** 

 Doc 2: Pure Moods Pure Moods was the first United States release of a series of compilation albums of new-age music released by Virgin Records. The original was titled "Moods – A contemporary Soundtrack" and released in the UK in 1991. This was followed by "Moods 2" in 1992. The series focuses on the genres of new-age, ambient, world music, and to a lesser extent, downtempo, trip-hop and smooth jazz. 

 Doc 8: Christina Lux, Alquimia and Gleisberg; and several compilations of meditative and world-influenced New Age music including "Musical Healing" in 2001 and "Perfect Balance" in 2006. In 2006, the Gordon brothers released "Soothing Sanctuary", ambient music with nature sounds, a re-version of the earlier release "Sanctuary" (so stated in the liner notes) with new arrangements and remastering. 

 Doc 14: number that expanded to 200 as of March 2010 with the release of "Shaman", a duo recording featuring Jean Derome and Joane Hétu. Presently DAME has a catalogue of over 450 titles. Ambiances Magnétiques began in 1982 when René Lussier, a guitarist, and Jean Derome, who plays saxophone and flute, performed live as a duo in Montreal under the name Ambiances Magnétiques 

 (Total 40 Documents) 

 **Response:** **Ambient** ✗ |
| DynamicRAG | **(Reranker Results:)** 

 Doc 5: over the fact that the album was not released as of May 2018, in the group's 2018 Mannheim Steamroller Christmas Tour announcement, founder Chip Davis announced that Exotic Spaces would be officially released upon the start of the tour, being sold at all concert locations as well as on www.mannheimsteamroller.com and Amazon.com. "Billboard"s Top New Age Albums chart became the New Age Albums chart in June 2009. Mannheim Steamroller is an American Neoclassical new-age music group founded by Chip Davis that is known primarily for its "Fresh Aire" series of albums, which blend classical music with elements of 

 Doc 17: Neoclassical new-age music content: Neoclassical new-age music Within the broad movement of new-age music, neoclassical new-age music, or instrumental pop, is influenced by and sometimes also based upon early, baroque or classical music, especially in terms of melody and composition. 

 Doc 3: to an increase CD sales and eventually awards with a Western Canadian Music Awards (WCMA) (best dance) and a Juno Award nomination (best instrumental). Canadian success and a trip to MIDEM in France led to a deal with Bay Area indie label XDOT25 which released the EP in the USA. 

 (Total 7 Documents) 

 **Response:** **Mannheim Steamroller** ✓ |
| Ground Truth | Mannheim Steamroller |

Table 11: Case Study for DynamicRAG. We compare our method with Vanilla-RAG with Reranker (monoT5). The initial retrieved content is the same for both methods. We take Top-40 documents as the input.

| Question | where is arachidonic acid found in the body |
|---|---|
| GPT-4o | **Cell Membranes, Brain, Liver, Platelets, Eicosanoid Production Sites.** |
| Vanilla-RAG w/Reranker | **(Reranker Results:)** 

 Doc 1: present in the phospholipids (especially phosphatidylethanolamine, phosphatidylcholine, and phosphatidylinositides) of membranes of the body's cells, and is abundant in the brain, muscles, and liver. Skeletal muscle is an especially active site of arachidonic acid retention, accounting for roughly 10-20% of the phospholipid fatty acid content typically. In addition to being involved in cellular signaling as a lipid second messenger involved in the regulation of signaling enzymes, such as PLC-, PLC, and PKC, and isoforms, arachidonic acid is a key inflammatory intermediate and can also act as a vasodilator. (Note separate synthetic pathways, as described in section below.) 

 Doc 3: Arachidonic acid Arachidonic acid (AA, sometimes ARA) is a polyunsaturated omega-6 fatty acid 20:4(-6), or 20:4(5,8,11,14). It is structurally related to the saturated arachidic acid found in cupuaçu butter ("L. arachis" – peanut). In chemical structure, arachidonic acid is a carboxylic acid with a 20-carbon chain and four "cis"-double bonds; the first double bond is located at the sixth carbon from the omega end. Some chemistry sources define árachidonic acidŕo designate any of the eicosatetraenoic acids. However, almost all writings in biology, medicine, and nutrition limit the term to all "cis"-5,8,11,14-eicosatetraenoic acid. Arachidonic acid is a polyunsaturated fatty acid... 

 Doc 5: arachidonic acid supplementation for Alzheimer's patients are needed. Another study indicates that air pollution is the source of inflammation and arachidonic acid metabolites promote the inflammation to signal the immune system of the cell damage. Arachidonic acid is marketed as an anabolic bodybuilding supplement in a variety of products. Supplementation of arachidonic acid... 

 (Total 40 Documents) 

 **Response: brain, muscles, and liver ✓** |
| DynamicRAG | **(Reranker Results:)** 

 ***None*** 

 (Total 0 Documents) 

 **Response: brain, muscles, liver ✓** |
| Ground Truth | brain, muscles and liver. |

Table 12: Prompt template for dynamic Reranker.

| Dynamic Reranker Prompt |
| --- |
| You are an expert at dynamically generating document identifiers to answer a given query. 
 I will provide you with a set of documents, each uniquely identified by a number within square brackets, e.g., [1], [2], etc. 
 Your task is to identify and generate only the identifiers of the documents that contain sufficient information to answer the query. 
 Stop generating identifiers as soon as the selected documents collectively provide enough information to answer the query. 
 If no documents are required to answer the query, output "None". 
 Output the identifiers as a comma-separated list, e.g., [1], [2] or "None" if no documents are needed. 
 Focus solely on providing the identifiers. Do not include any explanations, descriptions, or additional text. 
 *Query: { Question }* 
 *Retrieved Content:* 
 *1. Title: { title } Content: { content }* 
 *...* 
 *Top-N. Title: { title } Content: { content }* |

Table 13: Prompt template for retrieval-based generator.

| Retrieval-based Generator Prompt |
| --- |
| You are an intelligent assistant that uses retrieved knowledge to answer user queries accurately and concisely. Follow these rules: 

 1. **Task**: 
   • Use the provided `[Retrieved Content]` to generate responses. 
   • If the Retrieved Content is None, you should generate an answer based on your own knowledge. 
   • If the information is insufficient or you don't know the answer, state, "I cannot fully answer based on the available information. Please provide more details." 

 2. **Requirements**: 
   • **Accuracy**: Base your answers on the retrieved content. 
   • **Conciseness**: Keep answers brief and relevant. 
   • **Context Awareness**: Ensure your responses align with the user's query. 

 3. **Input Format**: 
   • Query: `[User Query]` 
   • Retrieved: `[Retrieved Content]` 

 4. **Output Format**: 
   • A structured, clear response tailored to the query. 

 Always prioritize clarity and reliability. 
 *Query: { Question }* 
 *Retrieved Content:* 
 *1. Title: { title } Content: { content }* 
 *...* 
 *Top-K. Title: { title } Content: { content }* |

Table 14: Prompt template for our designed reward function.

**Reward Function Prompt**

Use the following criteria to evaluate the quality of the model's response in a knowledge-intensive task, considering the provided ground-truth answer. Assign a score between 0-100 based on the overall quality, relevance, and correctness of the response:
1. Relevance to the Prompt (20 points):
Award up to 20 points if the response aligns well with the user's query, even if minor errors are present. Deduct points if the response lacks focus or deviates significantly from the query.
2. Accuracy of Factual Information (20 points):
Grant up to 20 points for correct factual details aligning with the ground-truth answer. Penalize for inaccuracies, missing essential elements, or presenting incorrect knowledge.
3. Handling of Temporal and Logical Reasoning (20 points):
Award up to 20 points for demonstrating correct temporal and logical reasoning. Deduct points if temporal reasoning is flawed or logical consistency is missing.
4. Clarity and Coherence of Response (20 points):
Assign up to 15 points for clear, coherent, and well-structured responses. Reduce points for ambiguity, confusion, or poor organization.
5. Potential Misleading Nature or Misconceptions (20 points):
Award up to 10 points if the response avoids being misleading. Penalize responses that could confuse or mislead the user, even if partially relevant. After evaluating the response based on these criteria, provide a total score in the format: "Score: points".
`Few-Shot Examples`
*User: { Instruction }*
*Ground-Truth Answer: { Answer }*
*Model Response: { response }*

Table 15: Prompt template for GPT baselines.

**GPT Baseline Prompt**

Below is a question, directly generate the answer. Your answer should be as concise as possible, which can be a word or a phrase.
*Question: { Instruction }*
*Response:*

Table 16: Prompt template for Llama baselines.

**Llama Baseline Prompt**

Below is an instruction that describes a task.
Write a response that appropriately completes the request.
*Paragraph: { Retrieved documents }*
*Question: { Instruction }*
*Response:*

