# OpenReview forum: "DynamicRAG: Leveraging Outputs of Large Language Model as Feedback for Dynamic Reranking in Retrieval-Augmented Generation"
_NeurIPS.cc/2025/Conference — NeurIPS 2025 poster_

### Official Review · Reviewer_mLW6 · 2025-06-16

**Clarity:** 3
**Significance:** 3
**Originality:** 3
**Rating:** 5
**Confidence:** 4

**Summary:**

This paper introduces DynamicRAG, a retrieval-augmented generation framework in which the reranker is treated as an agent that dynamically selects both the order and the number of retrieved documents for each query. Training proceeds in two phases: first, behavior cloning from expert trajectories provides a supervised initialization; second, reinforcement learning refines the reranker’s policy using multi-dimensional rewards derived from the quality of the LLM’s generated outputs.

**Questions:**

1. Have the authors considered applying DynamicRAG to tasks beyond question answering, and if so, what modifications might be necessary?
2. How robust is performance to the relative weights of the five reward components?

**Ethical Concerns:**

["NO or VERY MINOR ethics concerns only"]

**Final Justification:**

I have read the rebuttal, which has fully resolved my concerns. I decide to keep my score positive.

**Limitations:**

See weakness above

**Quality:**

4

**Strengths And Weaknesses:**

Strengths:
1. The idea of framing reranking as an interactive agent that learns from the generator’s feedback brings a principled integration of retrieval and generation objectives.
2. The two-stage training strategy leverages behavior cloning to bootstrap the policy and reinforcement learning to optimize for downstream quality, balancing sample efficiency and task alignment.
3. The reward design captures multiple facets of answer quality, from factual correctness to linguistic fluency, offering a rich supervisory signal.

Weaknesses:
1. The focus remains on question answering; it is unclear how readily the method extends to other retrieval-augmented generation tasks such as summarization or code generation.
2. While the efficiency comparison is promising, end-to-end latency numbers under realistic deployment conditions are not reported.

---

> ### Author Response · Authors · 2025-08-01
> **Rebuttal for Reviewer mLW6**
>
> Thank you for your positive feedback regarding "principled integration of retrieval and generation objectives", “balancing sample efficiency and task alignment” and "rich supervisory signal". We will address each point below to clarify and resolve your concerns:
>
> 1. "The focus remains on question answering": Indeed, one important scenario for RAG is open-domain QA problems. Many recent related works, such as Search O1 and Search R1, use Exact Match as the sole directly optimized reward and have achieved excellent results on QA tasks. We chose to incorporate other rewards, such as BERT-score, entirely to improve the robustness of our training. Additionally, we have demonstrated the effectiveness of our designed rewards on non-QA tasks, such as fact checking (FEVER) and long text generation (ELI5). Furthermore, we evaluated our model's performance on slot filling tasks (T-REx), as shown in the following table:
> | Model | T-REx |
> |-------|-------|
> | GPT-3.5-Turbo | 33.6 |
> | GPT-4 | 41.7 |
> | GPT-4o | 42.3 |
> | LLaMA2-7B | 22.3 |
> | LLaMA2-7B w/ Reranker | 25.6 |
> | LLaMA2-7B-SFT | 30.8 |
> | Ours (7B) | 41.5 |
> | LLaMA3-8B | 25.3 |
> | LLaMA3-8B w/ Reranker | 27.6 |
> | LLaMA3-8B-SFT | 42.4 |
> | Ours (8B) | **56.8** |
> As shown in the table, our 8B model outperforms GPT-4o on the T-REx dataset, demonstrating strong generalization capabilities beyond standard QA tasks.
>
> 2. "end-to-end latency numbers under realistic deployment conditions are not reported.": Thank you for pointing this out! In fact, our Table 4 already reports the specific runtime when deployed using vLLM on 8 A100 GPUs (we did not include retrieval time because this step is required for all methods). As can be seen, Vanilla-RAG takes approximately 0.57 seconds, LLM-based list-wise reranking takes approximately 0.75 seconds, while traditional reranking that concatenates question and document takes 13 seconds. Our method takes approximately 0.61 seconds, which is slightly slower than the Vanilla-RAG setting, but faster than all other methods.
>
> 3. “Have the authors considered applying DynamicRAG to tasks beyond question answering”: In addition to mainstream QA task datasets (NQ, TriviaQA), we also tested on the fact checking dataset (FEVER) in our main table. Furthermore, we tested on the slot filling task dataset (T-REx). The results can be found in the response to the first weakness.
> As can be seen, DynamicRAG is essentially an improvement to RAG, so it can be applied to all tasks that are compatible with RAG. We achieved excellent results on non-QA tasks without any modifications, which can be attributed to our consideration of non-QA rewards such as BertScore, Rouge, etc., when designing the reward functions.
>
>
> 4. "How robust is performance to the relative weights of the five reward components?": We detailed the impact of different reward designs on final results in Appendix Section C.3. As shown in Table 5, overall, the consistent trends across different reward function configurations highlight that the model's success is predominantly driven by the synergy of EM, SS, TF, and LP, with LLMEval serving as a supplementary component in refining performance. Additionally, we also tested the hyperparameters of these five rewards, as shown in Table 6. The results show that different parameter settings yield similar performance, indicating that our framework is robust and largely insensitive to these hyperparameters.
>
> Once again, thank you for your thorough review. I hope these points can address your concerns.

---

> ### Comment · Reviewer_mLW6 · 2025-08-06
>
> Thanks for your feedback! The rebuttal has fully resolved my questions and I decide to keep my score positive.

---

> ### Author Response · Authors · 2025-08-06
> **Response for Reviewer mLW6**
>
> Dear Reviewer mLW6,
>
> Thank you very much for your constructive feedback on our submitted paper. We were particularly grateful to see that you keep your positive score following our responses.
>
> We sincerely appreciate the time and effort you dedicated to reviewing our work. Your insightful comments were invaluable in helping us improve the manuscript.
>
> Thank you again for your support and positive evaluation.
>
> Best,
>
> Author

---

### Official Review · Reviewer_tviR · 2025-07-03

**Clarity:** 3
**Significance:** 2
**Originality:** 2
**Rating:** 4
**Confidence:** 4

**Summary:**

This paper introduces DynamicRAG, a RAG framework where a reranker is treated as an RL agent that dynamically selects both the order and number of retrieved documents per query. In this way, the irrelevant information in RAG could be mitigated and thus improve the overall performance. The proposed method includes two stages. First, some expert trajectories are used to cold-start the reranker via behavior cloning. Then, the reranker refines its policy via DPO over multiple sampled trajectories, with rewards derived from a multi-dimensional evaluation of the LLM’s generated output. Experiments are conducted on 7 datasets, focusing on llama3 8B and llama2 13B LLM backbones, demonstrating good empirical results over baselines.

**Questions:**

See above.

**Ethical Concerns:**

["NO or VERY MINOR ethics concerns only"]

**Final Justification:**

My major concerns were on the experiments on larger models, training costs, and hyperparameter sensitivity. From the response to me, I think my concerns have been resolved.

**Limitations:**

See above.

**Paper Formatting Concerns:**

No.

**Quality:**

2

**Strengths And Weaknesses:**

### Strengths
- The two-stage training idea is straightforward to improve RAG systems with irrelevant noise. The ablation studies verify the effectiveness of each component in DynamicRAG
- Leveraging LLMs' feedback as a supervision for reranking is interesting, and the use of DPO makes sense.
- The empirical results look good, with many popular datasets being included.

### Weaknesses
- Relaying on relatively small LLMs (LLaMA2-7B, 13B, LLaMA3-8B) for evaluation limits generality to larger, more popular models.
    - Have the authors tried DynamicRAG with larger LLMs (e.g., Llama-70B, GPT-4) or shown whether the proposed strategy still yields gains at scale?
    - In addition, have the authors tried reasoning models? Do they also suffer from the issue caused by irrelevant info? Can the proposed method also be paired with reasoning models and improve their performance?
- Engineering-heavy reward design (five weighted metrics) complicates reproducibility and may require extensive hyperparameter tuning.
    - How sensitive is DynamicRAG to the choice of weights in Eq.7?
- While Figure 4 highlights LLM call efficiency, what is the end-to-end training cost (compute hours) compared to static rerankers?

---

> ### Author Response · Authors · 2025-08-01
> **Rebuttal for Reviewer tviR**
>
> Thank you for your positive feedback regarding “ straightforward”, “ interesting”, "the use of DPO make sense" and "effectiveness". We will address each point below to clarify and resolve your concerns:
>
> 1. "Have the authors tried DynamicRAG with larger LLMs": Due to resource limitations, we cannot directly train models larger than 13B (such as LLaMA 70B). However, we conducted experiments with GPT-4o through API, as shown in Table 7 in section C.4. The results show that the Dynamic Reranker module consistently improves GPT-4o's performance across all datasets, with gains of 2.3, 0.8, and 0.7 percentage points on NQ, HotpotQA, and ASQA, respectively.
>
> 2. "In addition, have the authors tried reasoning models?": We tested QwQ-32B's performance on NQ and TriviaQA with different retrieval top-n settings, as shown in the following table:
> | Method | NQ | TriviaQA | Avg. |
> |--------|----|---------|----- |
> | QwQ (n=5) | 32.8 | 61.3 | 47.1 |
> | QwQ (n=10) | 34.0 | 63.4 | 48.7 |
> | QwQ (n=20) | 33.6 | 62.5 | 48.0 |
> We can observe that reasoning models still suffer from this problem: We observe a trade-off of selecting top-k contexts: a smaller k compromises the recall, while a larger k could introduce irrelevant or noisy context and mislead the LLM generation.
> Instead of training based on the QwQ foundation, we followed current mainstream practices by using Qwen2.5-3B-Instruct and employing the GRPO algorithm to let it explore reasoning chains with think tags (note that we only used Exact Match as the reward function at this stage). The results are as follows:
> | Method | NQ | TriviaQA | Avg. |
> | Qwen-2.5-3B-Instruct (RAG) | 34.8 | 54.4 | 44.6 |
> | with Reranker | 35.2 | 55.6 | 45.4 |
> | Ours | 37.6 | 58.7 | 48.2 |
>
> As can be seen, our method significantly improves performance, with an average improvement of approximately 3.6 points. This demonstrates that our approach achieves further improvements in reasoning mode.
>
> 3. "How sensitive is DynamicRAG to the choice of weights in Eq.7?": We detailed the impact of different reward designs on final results in Appendix Section C.3. As shown in Table 5, overall, the consistent trends across different reward function configurations highlight that the model's success is predominantly driven by the synergy of EM, SS, TF, and LP, with LLMEval serving as a supplementary component in refining performance. Additionally, we also tested the hyperparameters of these five rewards, as shown in Table 6. The results show that different parameter settings yield similar performance, indicating that our framework is robust and largely insensitive to these hyperparameters.
>
> 4. "what is the end-to-end training cost (compute hours) compared to static rerankers?": For example, regarding RankLLaMA (static reranker), its paper clearly states that it uses 16 V100 GPUs to train a 4k context length reranker for 8 days. In contrast, we need 8 A100 GPUs to train a dynamic reranker based on LLaMA2-7B (also with 4k context length) for 2 days. Therefore, the specific gpu-hours ratio is 3072 gpu-hours on V100 versus 384 gpu-hours on A100. However, considering that A100's actual computational power is approximately 2.5-3 times that of V100, the final cost ratio may be 3:1, meaning our training cost is one-third of theirs.
>
> Once again, thank you for your thorough review. I hope these points can address your concerns.

---

> ### Author Response · Authors · 2025-08-06
> **Kind Request for Timely Feedback on our Responses**
>
> Dear Reviewer tviR,
>
> We are writing to follow up on the response we submitted regarding your comments.
>
> Firstly, we would like to express my gratitude for your initial review and valuable feedback. The insights provided have been instrumental in refining our paper, and we have tried our best to address each point in our response with the aim of enhancing the paper's quality and relevance.
>
> Understanding the time and effort required for a thorough review, we greatly appreciate the commitment you have shown towards ensuring the high standards of NeurIPS. However, as the rebuttal process is concluding, we are keenly awaiting your further feedback on our responses. The comments from your expertise are crucial for the final preparation and improvement of our paper.
>
> If our responses more closely meet your expectations for the paper, we respectfully ask you to reconsider your initial rating. If you have any further questions or require more information to raise your initial score, please feel free to let us know. We are fully committed to making all necessary adjustments to meet your expectation.
>
> Thank you once again for your time and dedication. We look forward to your valuable feedback.
>
> Author

---

> > ### Comment · Reviewer_tviR · 2025-08-08
> >
> > I thank the authors for their detailed response, which has resolved most of my concerns, and thus, I have increased my rating.

---

> > > ### Author Response · Authors · 2025-08-08
> > > **Reply for Reviewer tviR**
> > >
> > > Dear Reviewer tviR,
> > >
> > > Thank you very much for your constructive feedback on our submitted paper. We were particularly grateful to see that you raised your score following our responses.
> > >
> > > We sincerely appreciate the time and effort you dedicated to reviewing our work. Your insightful comments were invaluable in helping us improve the manuscript.
> > >
> > > Thank you again for your support and positive evaluation.
> > >
> > > Best,
> > >
> > > Author of DynamicRAG

---

### Official Review · Reviewer_Mhs5 · 2025-07-03

**Clarity:** 3
**Significance:** 2
**Originality:** 4
**Rating:** 4
**Confidence:** 3

**Summary:**

This paper addresses a core limitation of traditional RAG rerankers—the use of a fixed number of retrieved documents (k), which often fails to balance the trade-off between information loss (when k is too small) and noise (when k is too large). To overcome this, the authors propose DynamicRAG, a novel framework that jointly optimizes both the number of retrieved documents and their ranking order. Built upon the SFT+DPO training paradigm, DynamicRAG significantly improves reranker performance, achieving higher generation quality with less training data compared to strong baselines.

**Questions:**

In addition to the concerns raised under Weaknesses, I have two further questions:
1. A significant portion of the main experimental results appears to be missing. Is there a specific reason for this?
2. What is the rationale behind the choice of hyper-parameters? Additionally, how sensitive is the method to these hyper-parameters? Asking because the cost of tuning them on different datasets could be substantial and non-negligible. Although Appendix C.3 lists three hyper-parameters, is there any theoretical or empirical justification for their selection?

**Ethical Concerns:**

["NO or VERY MINOR ethics concerns only"]

**Limitations:**

Yes

**Quality:**

3

**Strengths And Weaknesses:**

Strength:
1. Clear motivation: The authors highlight that relying on a fixed number of retrieved documents (k) inherently fails to balance the trade-off between information loss (when k is too small) and noise introduction (when k is too large). This is a valuable and often overlooked insight in existing research.
2. Targeted method design: The paper proposes a dynamic reranking mechanism built on a reinforcement learning framework, allowing the reranker to adaptively adjust both the number and order of retrieved documents based on feedback from LLM-generated outputs.
3. Strong performance validation: Extensive experiments on seven knowledge-intensive datasets demonstrate that DynamicRAG achieves state-of-the-art performance on most benchmarks, while maintaining high efficiency and consistency across different backbone sizes. The comprehensive experimental setup and reproducibility of results further strengthen the credibility of the findings.

Weaknesses：
1. The core premise of the paper is that dynamic determination of k  enhances generation quality, yet the ablation experiments lack explicit validation of k's impact. For instance, fixing k during RL training to isolate its effect—e.g., comparing performance with static k vs. dynamic k —would better demonstrate the value of adaptive selection.
2. The paper lacks a direct comparison with RankRAG, despite their similar focus and comparable performance. Though Appendix C.5 shows the trend of DynamicRAG’s performance, it does not substitute for direct comparative experiments—e.g., scaling DynamicRAG to 470k data or reducing RankRAG to 150k—to conclusively demonstrate efficiency advantages.
3. Key definitions in the framework require clarification:
(i) In Figure 3, the threshold condition "s_i > T“ for document filtering lacks explanation—including how T is set and handling cases where no documents meet the threshold.
(ii) The "expert model" used to score documents in Figure 3 is undefined. Whether it relies on human annotations, LLM evaluations, or another metric (e.g., MonoT5 reranking) and its reliability criteria are unspecified, leaving the training signal's validity ambiguous. These omissions obscure the framework's technical details for readers.

---

> ### Author Response · Authors · 2025-08-01
> **Rebuttal for Reviewer Mhs5**
>
> Thank you for your positive feedback regarding "the clear motivation on balancing information loss and noise introduction", "the targeted method design with dynamic reranking and RL adaptation", and "the strong performance validation across diverse benchmarks". We will address each point below to clarify and resolve your concerns:
>
> 1. "static k vs. dynamic k": We conducted the following experiment: we processed all samples in the training set with k=12 and k≥12 (the average number of k after RL training with non-fixed k is 12), and then performed training. The results are shown in the following table:
>
> | Method | NQ | HotpotQA | ASQA | Avg |
> |--------|----|---------|----- |-----|
> | VanillaRAG | 39.1 | 31.5 | 46.8 | 39.1 |
> | DynamicRAG (fixed k=12) | 45.3 | 35.1 | 50.8 | 43.7 |
> | DynamicRAG (dynamic k) | 48.4 | 36.7 | 56.3 | 47.1 |
> The results show that training with a fixed k value performs better than normal reranking, but worse compared to the non-fixed k setting. We believe this is consistent with our hypothesis that different problems require different k values, while fixed k both increases irrelevant information interference to some extent and increases the possibility of insufficient information.
>
> 2. "direct comparison with RankRAG": Thank you for pointing this out! RankRAG is an important baseline in this paper, but since its source code has not been made publicly available, we cannot reproduce its results. Consequently, we are unable to reduce RankRAG's training data to 150k for training and comparison purposes. One of our key contributions is training a better and more comprehensive RAG system with less data, and we conducted data scaling experiments in Table C.5 to verify that further increasing data can improve the effectiveness of our method.
>
> 3. “Key definitions“: Thank you for pointing this out! Regarding ""s_i > T" for document filtering lacks explanation," we indeed did not explicitly specify its value in Figure 3's caption. In fact, we included this information in line 720 of section D.1, and we will add it earlier in the updated version of the paper to reduce misunderstanding. Additionally, if no document satisfies this condition, we discard that sample.
>
> Once again, we apologize for not clarifying this point. In fact, we mentioned in line 719 of section D.1 that we use MonoT5 as our expert model, and we take the probability of the "True" token from the probability distribution of its first output token as s_i. (MonoT5's output is either True or False).
>
> We will address both of these points in the updated version of the paper. Thank you again for your feedback and suggestions.
>
> Q1: "main experimental results appears to be missing": All baseline results in our main table are directly copied from their respective papers. Sometimes other baselines may not have evaluated on the datasets or benchmarks we selected, and some methods are difficult to reproduce, which leads to missing entries in the main table.
>
> Q2: "What is the rationale behind the choice of hyper-parameters?": All experiments in the main table are conducted based on hyperparameters (0.2, 0.2, 0.2, 0.2, 0.2), and this hyperparameter choice was made purely for simplicity. We conducted experiments in Appendix C.4 to explicitly demonstrate the robustness of hyperparameter selection in our framework. If you need any theoretical or empirical justification, we can provide a simple explanation: As shown in Table 5, after removing Exact Match, the final model's performance degrades significantly more compared to removing other reward functions. Many other works, such as Search O1 and Search R1, use Exact Match as the sole directly optimized reward and achieve excellent results on QA tasks. This also indicates that Exact Match serves as the dominant reward, while other rewards play auxiliary roles.
>
> Once again, thank you for your thorough review. I hope these points can address your concerns.

---

> ### Author Response · Authors · 2025-08-06
> **Kind Request for Timely Feedback on our Responses**
>
> Dear Reviewer Mhs5,
>
> We are writing to follow up on the response we submitted regarding your comments.
>
> Firstly, we would like to express my gratitude for your initial review and valuable feedback. The insights provided have been instrumental in refining our paper, and we have tried our best to address each point in our response with the aim of enhancing the paper's quality and relevance.
>
> Understanding the time and effort required for a thorough review, we greatly appreciate the commitment you have shown towards ensuring the high standards of NeurIPS. However, as the rebuttal process is concluding, we are keenly awaiting your further feedback on our responses. The comments from your expertise are crucial for the final preparation and improvement of our paper.
>
> If our responses more closely meet your expectations for the paper, we respectfully ask you to reconsider your initial rating. If you have any further questions or require more information to raise your initial score, please feel free to let us know. We are fully committed to making all necessary adjustments to meet your expectation.
>
> Thank you once again for your time and dedication. We look forward to your valuable feedback.
>
> Author

---

### Official Review · Reviewer_VGiT · 2025-07-03

**Clarity:** 2
**Significance:** 3
**Originality:** 2
**Rating:** 4
**Confidence:** 3

**Summary:**

DynamicRAG enhances a standard retriever by adding a dynamic reranker that adaptively selects both the order and the number kk of documents retrieved for each query. This reranker is initially trained via supervised fine-tuning to mimic expert behavior and then optimized through reinforcement learning using Direct Preference Optimization (DPO), treating the language model as the environment. The reward signal integrates multiple components, including Exact Match, BERTScore, ROUGE, length penalties, and LLM-based evaluations, with ablation studies revealing that Exact Match is especially influential. Using LLaMA-3-8B, the model achieves state-of-the-art results for its size, outperforming even GPT-4o in a non-retrieval setting.

**Questions:**

Which generator-independent reward signals could substitute for, or complement, Exact Match so that DynamicRAG remains reliable on tasks where the correct answer cannot be determined by straightforward token overlap?

**Ethical Concerns:**

["NO or VERY MINOR ethics concerns only"]

**Final Justification:**

Overall, I find the proposed approach to be practical and well-motivated. However, since my concerns have not been fully resolved, I am assigning a weak accept.

**Limitations:**

Please refer to the weaknesses.

**Paper Formatting Concerns:**

None.

**Quality:**

2

**Strengths And Weaknesses:**

**Strengths:** DynamicRAG adapts the passage budget *k* to the difficulty of each query, trimming irrelevant documents and spotlighting the most salient evidence. Its reinforcement-learning training optimises directly for answer quality, making the system more robust than fixed-threshold or static-score methods and less sensitive to hyperparameter tuning.

**Weaknesses:** DynamicRAG’s reward is computed directly from the generator’s output, so the reward distribution itself shifts whenever the generator is replaced or its decoding parameters are altered. For instance, coupling the trained reranker with a stronger external LLM such as GPT-4o yields an immediate jump in EM score, yet attaining peak performance still demands RL fine-tuning tailored to that new pairing. This reveals that the current reward scheme is overly tied to a particular generator and lacks true model-agnostic generality. Moreover, although the framework nominally combines five reward components, Exact Match is overwhelmingly dominant: removing it lowers average performance by more than eleven points. Consequently, the system is effectively optimised for token-level answer overlap, and its robustness on tasks requiring long chain-of-thought reasoning, open-ended generation, or other settings where correctness cannot be judged by simple string matching remains unverified.

---

> ### Author Response · Authors · 2025-08-01
> **Rebuttal for Reviewer VGiT**
>
> Thank you for your positive feedback regarding “making the system more robust”, and "less sensitive to hyperparameter tuning". We will address each point below to clarify and resolve your concerns:
>
> 1. We conducted the following experiments, applying the currently best-performing reranker (LLaMA3-8B) and a relatively weaker one (LLaMA2-7B) to other generators. The results are as follows:
> | Reranker | Generator | NQ (EM) | HotpotQA (EM) | ASQA (EM) | Avg |
> |----------|-----------|---------|---------------|-----------|-----|
> | LLaMA2   | LLaMA2    | 38.7    | 29.4          | 41.1      | 36.4|
> | LLaMA2   | LLaMA3    | 39.4    | 30.6          | 43.5      | 37.8|
> | LLaMA3   | LLaMA2    | 41.6    | 32.7          | 47.3      | 40.5|
> | LLaMA3   | LLaMA3    | 48.4    | 36.7          | 56.3      | 47.1|
> The results demonstrate that using a reranker trained on LLaMA2 combined with a generator trained on LLaMA3 yields better performance than using both reranker and generator trained solely on LLaMA2, which can be attributed to the superior performance of the generator. Furthermore, when using a reranker trained on LLaMA3 paired with a generator trained on LLaMA2, the performance is even better, indicating that the effectiveness of retrieval determines the generator's performance. Therefore, we argue that there is no "lacks true model-agnostic generality".
>
> 2. "The system has been effectively optimized for token-level answer overlap": Indeed, one important scenario for RAG is open-domain QA problems. Many recent related works, such as Search O1 and Search R1, use Exact Match as the sole directly optimized reward and have achieved excellent results on QA tasks. We chose to incorporate other rewards, such as BERT-score, entirely to improve the robustness of our training. Additionally, we have demonstrated the effectiveness of our designed rewards on non-QA tasks, such as fact checking (FEVER) and long text generation (ELI5). Furthermore, we evaluated our model's performance on slot filling tasks (T-REx), as shown in the following table:
>
> | Model | T-REx |
> |-------|-------|
> | GPT-3.5-Turbo | 33.6 |
> | GPT-4 | 41.7 |
> | GPT-4o | 42.3 |
> | LLaMA2-7B | 22.3 |
> | LLaMA2-7B w/ Reranker | 25.6 |
> | LLaMA2-7B-SFT | 30.8 |
> | Ours (7B) | 41.5 |
> | LLaMA3-8B | 25.3 |
> | LLaMA3-8B w/ Reranker | 27.6 |
> | LLaMA3-8B-SFT | 42.4 |
> | Ours (8B) | **56.8** |
> As shown in the table, our 8B model outperforms GPT-4o on the T-REx dataset, demonstrating strong generalization capabilities beyond standard QA tasks.
>
> Once again, thank you for your thorough review. I hope these address your concerns.

---

> > ### Comment · Reviewer_VGiT · 2025-08-05
> >
> > Thank you for addressing my concerns in a thorough and thoughtful manner. That said, I still have some remaining reservations regarding *reward distribution shift*. Since the reward is computed based on the generator’s output, there is a possibility that the reward distribution may vary depending on the decoding strategy (e.g., greedy vs. sampling). While the authors have demonstrated a certain degree of generality through experiments involving different generator pairings, the underlying issue that the reward may be sensitive to generation dynamics still remains.
> >
> > Nevertheless, I do not consider this a critical flaw. Given the consistent performance improvements and generalization across diverse tasks, I believe the proposed system is sufficiently robust in practice. Therefore, despite this remaining concern, I am raising my score.

---

> > > ### Author Response · Authors · 2025-08-05
> > > **Reply for Reviewer VGiT**
> > >
> > > Dear Reviewer VGiT,
> > >
> > > Thank you very much for your constructive feedback on our submitted paper. We were particularly grateful to see that you raised your score following our responses.
> > >
> > > We sincerely appreciate the time and effort you dedicated to reviewing our work. Your insightful comments were invaluable in helping us improve the manuscript.
> > >
> > > Thank you again for your support and positive evaluation.
> > >
> > > Best,
> > >
> > > Author

---

### Note · Authors · 2025-08-12

We thank the reviewers for their valuable feedback and AC's time and assistance. We believe our rebuttal has addressed the main concerns and highlight the key contribution of our work: Dynamic optimization of both document selection and ranking in RAG through reinforcement learning with LLM feedback.

We have addressed all concerns through clarifications and additional experiments. Notably, two reviewers (VGiT, tviR) have increased their scores following our responses, while the other two reviewers (Mhs5, mLW6) have maintained their positive ratings of 4 and 5 respectively. Specifically, the primary concerns raised by reviewers include model-agnostic generality and reward distribution sensitivity (VGiT), validation of dynamic vs. static k and technical definitions (Mhs5), scalability to larger models and reasoning tasks (tviR), and extension beyond question answering tasks (mLW6). We demonstrated cross-model generalizability through comprehensive experiments with different generator-reranker pairings, conducted rigorous ablation studies comparing fixed vs. dynamic k settings, and validated effectiveness on larger models such as GPT-4o and reasoning models like QwQ-32B and Qwen2.5-3B. We additionally performed experiments on diverse tasks including fact-checking, slot-filling, and long-form generation, showing our method's broad applicability beyond QA tasks and confirming robustness across different reward weight configurations.

Our work fundamentally addresses why fixed document selection fails in RAG systems and provides a principled solution through dynamic adaptation. The major contribution of our work lies in three folds:

- Identifying a critical limitation that fixed-k retrieval inherently fails to balance the trade-off between information loss and noise introduction across different query complexities.

- Providing a comprehensive framework that treats reranking as an RL problem, using multi-dimensional LLM feedback as rewards to optimize both document selection and ranking simultaneously.

- Demonstrating practical effectiveness with state-of-the-art results across seven knowledge-intensive datasets while achieving superior training efficiency.

Our work particularly highlights a previously overlooked but critical aspect of rerankers-the dynamic determination of document quantity-which we believe will inspire important future discussions in the community about adaptive retrieval strategies and pave the way for more intelligent RAG system designs.

---

### Decision · Program_Chairs · 2025-09-17

**Decision:**

Accept (poster)

**Comment:**

This paper introduces DynamicRAG. The approach treats the RAG system as a RL agent. The system dynamically selects the optimal number (and order) of retrieved documents using the model response quality as a reward signal.

The reviewers noted the following strengths of the paper:
* The paper addresses a critical limitation (i.e., fixed choice of k) in modern RAG systems
* Framing the reranker as an RL agent that learns from e2e task performance is novel and well-motivated
* The method demonstrates SOTA across a wide range of datasets and model backbones, outperforming competitive baselines with less training data
* The list-wise reranking strategy is more efficient than commonly used point-wise reranking methods

The reviewers also identified a number of weakness with the paper, including:
* The initial version of the paper lacked a direct ablation study comparing a dynamic k strategy vs. a fixed k strategy.
* Reviewers initially raised concerns about the method's generalizability to SOTA closed models
* The complexity of the reward function raised questions about its robustness
* The initial focus on QA tasks made it unclear how well the framework might extend to other RAG applications

The authors' rebuttal was exemplary. It included new experiments that addressed the key weaknesses identified by the reviewers. This included a direct comparison between a dynamic and static k selection strategy, evals using SOTA frontier models like GPT-4o, and experiments on new tasks like slot-filling, all of which supported (and strengthened) the paper's claims. As a result of the strong rebuttal, two reviewers raised their scores and a third kept their already positive rating. This is a nice, well-executed paper that should be accepted for publication.